# Semantic Decomposition and Anomaly Detection of Tympanic Membrane Endoscopic Images

Dahye Song [1,†] , In Sik Song [2,†] , Jaeyoung Kim [3] , June Choi [2,*,‡] and Yeonjoon Lee [1,*,‡]

1  Department of Applied Artificial Intelligence, Hanyang University, Ansan 15208, Republic of Korea
2  Department of Otorhinolaryngology-Head and Neck Surgery, Ansan Hospital,
   Korea University College of Medicine, Ansan 15208, Republic of Korea
3  Core Research & Development Center, Korea University Ansan Hospital, Ansan 15208, Republic of Korea
*  Correspondence: mednlaw@korea.ac.kr (J.C.); yeonjoonlee@hanyang.ac.kr (Y.L.)
†  These authors contributed equally to this work.
‡  These authors contributed equally to this work.

**Abstract:** With the recent development of deep learning, the supervised learning method has been widely applied in otolaryngology. However, its application in real-world clinical settings is difficult because of the inapplicability outside the learning area of the model and difficulty in data collection due to privacy concerns. To solve these limitations, we studied anomaly detection, the task of identifying sample data that do not match the overall data distribution with the Variational Autoencoder (VAE), an unsupervised learning model. However, the VAE makes it difficult to learn complex data, such as tympanic membrane endoscopic images. Accordingly, we preprocess tympanic membrane images using Adaptive Histogram Equalization (AHE) and Canny edge detection for effective anomaly detection. We then had the VAE learn preprocessed data for only normal tympanic membranes and VAE was used to calculate an abnormality score for those differences between the distribution of the normal and abnormal tympanic membrane images. The abnormality score was applied to the K-nearest Neighbor (K-NN) algorithm to classify normal and abnormal tympanic membranes. As a result, we were obtained a total of 1232 normal and abnormal eardrum images, classified with an accuracy of 94.5% using an algorithm that applied only normal tympanic membrane images. Consequently, we propose that unsupervised-learning-based anomaly detection of the tympanic membrane can solve the limitations of existing supervised learning methods.

**Keywords:** VAE; anomaly detection; healthcare; tympanic membrane; otoscope images

## 1. Introduction

Otitis Media (OM) is one of the most common disorders in children worldwide and studies indicate that over 80% of infants up to the age of three might have OM, with many of these cases likely to recur [1,2]. Specifically, cholesteatoma is an ear lesion caused by the abnormal collection of skin cells, which gradually expands, resulting in the erosion of neighboring bone structures. It can cause various complications, including hearing loss, vestibular dysfunction, and facial paralysis. The causes of the disease vary, including perforation and retraction. Misdiagnosing middle ear disease can result in undertreatment or overtreatment, resulting in severe adverse consequences [3]. Additionally, many OM patients have been prescribed antibiotics, which increase resistance and, therefore, require a precise diagnosis. However, the average diagnostic accuracy of pediatricians and otolaryngologists is 50% and 73%, respectively [4].

Moreover, otolaryngology diagnosis is conducted by a visual mechanism for the otoscope. Therefore, diagnosis and precision may be biased, depending on the person diagnosing [5]. Therefore, using Artificial Intelligence (AI) in otolaryngology is reasonable. Recently, with the development of deep learning, more research has used AI technology for medical data [6] and more studies on middle ear disorders have been conducted [7]. Khan

et al. classified 2484 endoscopic images as normal, Chronic Otitis Media (COM), and OM with Effusion (OME) with 94.9% accuracy using DensNet161 [8]. Wu et al. trained Xception and MobileNet-V2 on 10,703 endoscopic images and classified normal, Acute Otitis Media (AOM), and OME, achieving an accuracy of 97.45% and 95.72%, respectively [9]. Eroğlu et al. used AlexNet, GoogLeNet, and DenseNet201 to extract features from 3093 temporal bone CT images. Then, they utilized a Support Vector Machine (SVM) to classify COM with cholesteatoma and without cholesteatoma with a 95.4% accuracy [10]. Thus, prior research has demonstrated good performance, with most of this based on supervised learning models. However, applying supervised learning algorithms to real-world clinical situations is difficult because of their inapplicability outside the training domain and bias due to imbalanced data [11]. To prevent this, an adequate amount of data (approximately 1000 data points per case) is necessary for all patient categories. However, privacy-related legislation limits the gathering and accessing of all types of data [12]. Creating a database that includes any type of data may result in issues, such as class imbalance, huge costs, and missing data [13].

To overcome the limitations of prior research, we propose a method that learns only normal tympanic membrane images using a VAE and calculates the distance between normal and abnormal data to detect abnormal data. Unlike in previous research, we only use normal data for learning; therefore, a large amount of data is unnecessary. In addition, the distance from the normal class is used as a classification criterion, allowing for the detection of data that cannot be collected or are difficult to diagnose with a specific condition or disease, despite being clearly abnormal.

However, tympanic membrane endoscopic data may be difficult for a model to comprehend because of their high complexity. Processing high-dimensional data to facilitate model analysis is a crucial issue. A previous study decomposed high-dimensional images semantically. Guo et al. estimated, with an AUC of 0.969, single RGB image 3D hand pose data [14]. The data were decomposed into independent factors, such as hand pose, shape, and color/texture, and applied to VAE. Tympanic membrane endoscopic images are also RGB images containing lots of information, such as tympanic membrane transparency and width and depth of the affected area. We applied AHE and Canny edge detection on endoscopic data to construct images incorporating tympanic membrane transparency and depth information and images containing the perforation width and inflammatory distribution to assist the model in understanding the data. Both types of images were converted into grayscale and learned from the model alongside the original RGB image. Anomaly detection is a significant topic in healthcare that has been extensively studied, for which many approaches have been presented.

Anomaly detection is the identification of data samples that do not conform to the distribution of all data (normal data) [15]. The VAE used in this study for anomaly detection is a generative model based on unsupervised learning [16]. Test data were fed into the VAE and it learned only normal data distribution. The difference between the learned image and the test image was calculated into an abnormality score, which was then applied by the machine learning classifier. The abnormality score was computed using the VAE reconstruction error and Kullback–Leibler diversity values.

In summary, this study proposes a method for overcoming the limitations of supervised-learning-based studies by detecting anomalies using a VAE after the semantic decomposition of high-complexity data through image preprocessing. Using this solution, we detected abnormalities in tympanic membrane endoscopic data with an accuracy of 94.5%. Our first contribution is the use of the semantic decomposition of highly complex data based on clinical evidence to facilitate the models' interpretation of images and improve anomaly detection performance. Our second contribution is that we can identify the abnormal findings of the tympanic membrane, including perforation, retraction, cholesteatoma, and certainly abnormal data, that are difficult to diagnose for a specific disease, analyzing only normal data.

The remainder of this paper is organized as follows. Section 2 briefly introduces the datasets and images used in this study, semantic decomposition, and anomaly detection. Sections 3 and 4 describe the image processing and abnormality score extraction procedures. We present the experiment results in Section 5 and discuss them in Section 6; Section 7 concludes the paper.

## 2. Materials and Methods

This section describes the datasets used in this study, collection procedures, and application methods. Then, we present a semantic decomposition of high-complexity data and describe how the VAE obtains the abnormality scores.

Tympanic membrane endoscopic data were gathered at the Korea University Ansan Hospital. In total, 1632 images were collected, comprising 469 normal, 558 perfection, 313 retraction, 159 cholesteatoma data points, and 133 images that were abnormal but difficult to classify as a specific disease. A total of 400 normal data points were used as learning data and the remaining 1232 data points were used to evaluate whether abnormal data were successfully detected using the abnormality scores calculated using the model. The IRB (2021AS0329) at the Korea University Ansan Hospital approved the retrospective data analysis. All procedures in this study adhered to the principles outlined in the 1975 Helsinki Declaration.

### 2.1. Semantic Decomposition of Tympanic Membrane Endoscopic Data

In this study, the semantic decomposition of tympanic membrane endoscopic images was based on clinical features. Previous studies have analyzed intricate images based on clinical features using image processing. Myburgh et al. used 389 tympanic membrane endoscopic images to detect perforation, fluid, mallei, etc.; vectorized the features of normal and foreign objects, AOM, OME, and CSOM; and used a decision tree to achieve an accuracy of 81.58% [17]. Existing research has linked the semantic decomposition of complicated images to the vectorization of clinical features; however, our study emphasizes clinical features via preprocessing and includes each in images, having directly learned from the model.

Three images were decomposed: the transparency and depth of the tympanic membrane, the range of perforation and inflammation, and the original RGB data. First, after converting the images to grayscale, AHE was applied to enhance image contrast such that only information regarding the tympanic membrane's transparency and depth of the afflicted area could be included. Subsequently, we used Canny edge detection to verify that only impacted edges remained, such as perfection and inflammation. Because both prior datasets were converted into grayscale, we combined the original datasets to retain color information, such as inflammation and pus. Anomaly detection was performed by extracting the abnormality scores for each of the three datasets.

### 2.2. Anomaly Detection

Anomaly detection is the process of recognizing samples that deviate from a learned data distribution. That is, it identifies outliers in a dataset that deviate from the norm [14]. While several algorithms for anomaly detection have been developed in the field of medical applications, Choi et al. trained a VAE on 353 brain Positron Emission Tomography (PET) data points and diagnosed patients with aberrant symptoms with an AUC of 0.74 [13]. PET data were used without additional preprocessing to detect anomalies with a single abnormality score. By contrast, we partitioned the data semantically and derived two abnormality scores from the model for detection.

First, the model learns only normal classes from the three decomposed images. The optimizer applied to train was Adam and the learning rate was learned by 100 epochs at $1 \times 10^{-3}$. The two types of abnormality scores were calculated by inserting test data into models trained only in the normal class, which then calculated the scores. Unlike Choi et al., who used only reconstruction errors as an abnormality score, we included

the Kullback–Leibler diversity as an abnormality score. Applying the KNN algorithm to the abnormality scores of each extracted dataset led to the detection of abnormal data containing the class that specialists struggle to classify as a particular disease but is certainly abnormal. Figure 1 shows the methodologies used in this study.

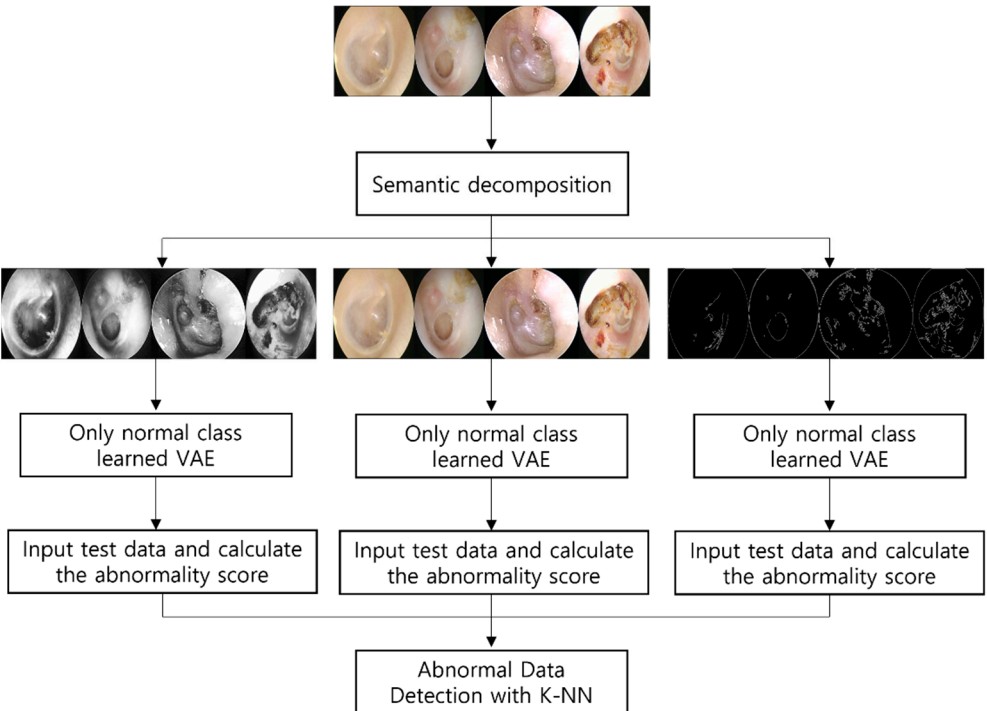

**Figure 1.** Overview of semantic decomposition of endoscopic data and anomaly detection methods using VAE.

## 3. Semantic Decomposition

Tympanic membrane endoscopic images comprise plenty of information, including the degree of inflammatory distribution, transparency of the tympanic membrane, presence of fluid, and degree of perforation. Therefore, semantically decomposing this information so the model can better comprehend it is vital. This section describes the techniques used for the semantic segmentation of tympanic membrane endoscopic images and the implications of each dataset. First, AHE highlights the tympanic membrane transparency and structural depth data. Then Canny edge detection is used to emphasize the degree and size of inflammation and perforation distribution.

### 3.1. Adaptive Histogram Equalization

In general, grayscale data, such as those in the MNIST dataset, perform better than RGB data when used in deep learning models. RGB images include a wider variety of information than grayscale images. Hsu et al. recognized polyps in colonoscopy images; however, their detection was more precise in grayscale images than in RGB/NBI images [18]. With the exception of complicated RGB data, we attempted to construct images that had only structural information, such as the depth of the damaged region and transparency of the tympanic membrane. After converting tympanic membrane endoscopic data into grayscale, the structure and depth of the tympanic membrane were highlighted using AHE. Figure 2 shows the original data with RGB values before applying AHE.

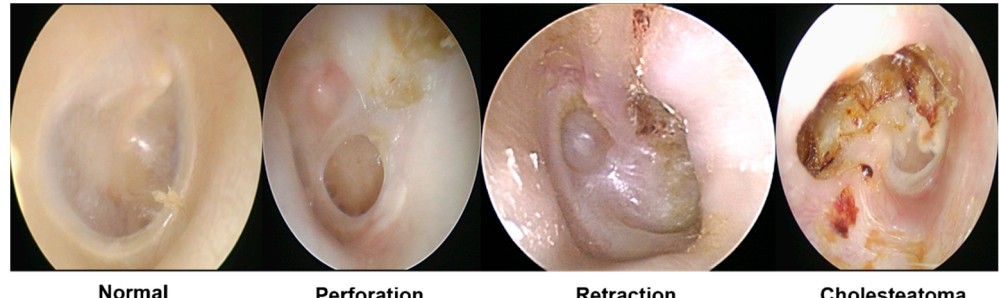

**Figure 2.** Original image before applying adaptive histogram equalization.

Histogram equalization (HE) achieves image equalization using a single histogram. If sections have distinct pixel intensity distributions and the image is equalized using a single histogram, the image may become deformed. Depending on the light utilized for photography, certain regions of the endoscopic images of the tympanic membrane may have high-intensity measures. AHE breaks images into numerous grids and equalizes them using histograms on each grid such that they can be processed without being distorted by the high-intensity regions caused by the reflection of light from the tympanic membrane endoscopic image. Consequently, we used AHE to prevent data loss owing to data distortion. Figures 3 and 4 show images of applying AHE.

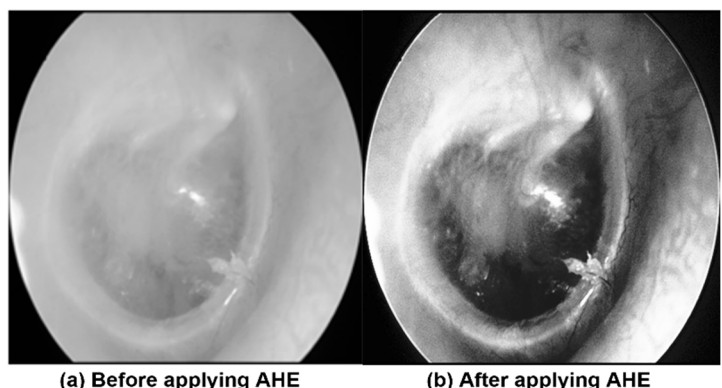

**Figure 3.** Before and after applying AHE.

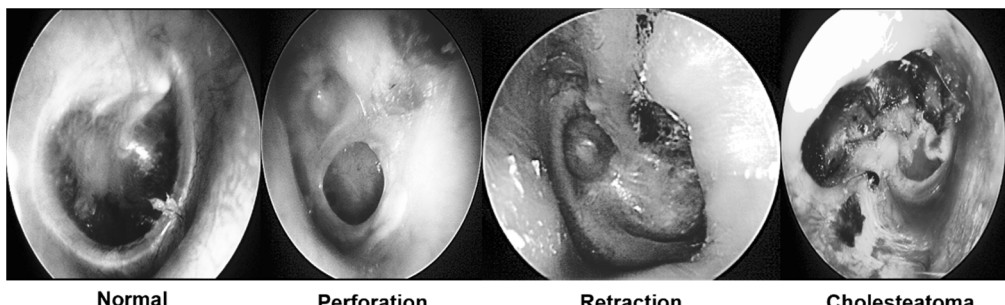

**Figure 4.** Image after applying adaptive histogram equalization.

### 3.2. Canny Edge Detection

Edges in an image are curves that indicate the boundaries of items and include crucial information, such as the shape and position of structures. Edge detection can preserve crucial properties of objects during image processing while filtering irrelevant information [19]. Numerous studies have been conducted on edge detection using medical data. For instance, Srinivas et al. detected images on an X-ray image and used Online Dictionary Learning (ODL) to classify body regions with 98.5% accuracy [20]. We applied Canny edge

detection to eardrum endoscopic images to filter only affected edges, which contain the most important information for anomaly detection. This edge-detection method can filter only important information from complex medical data and provide high-classification performances.

Canny edge detection is the most frequently used approach for edge detection [21]. First, noise is eliminated with a gaussian filter to produce a smooth image. Then, the intensity gradient in the smooth image is computed and nonmaximal suppression is employed to isolate the maximum gradient value. As noise can weaken edges after nonmaximal suppression, the maximum and minimum values are computed using a double threshold and only the maximum values are retained [22]. Because the features of endoscopic images are highly complex, the images in Section 3.1 were treated twice with gaussian filters before applying the Canny edge detection method to reduce noise. The minimum value was set to 10 and the maximum value was set to 250 to retain only the border of the affected region in the endoscopic image. Figures 5 and 6 depict the outcomes of applying Canny edge detection to each class.

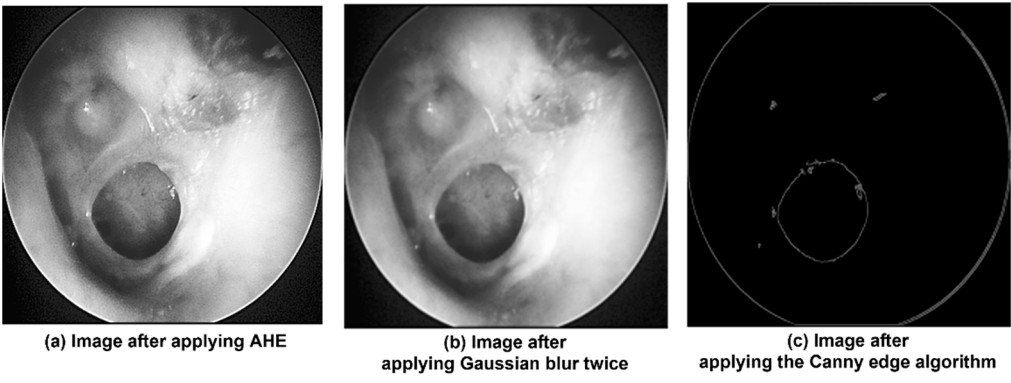

(a) Image after applying AHE  (b) Image after applying Gaussian blur twice  (c) Image after applying the Canny edge algorithm

**Figure 5.** Process of applying Canny edge-detection algorithm.

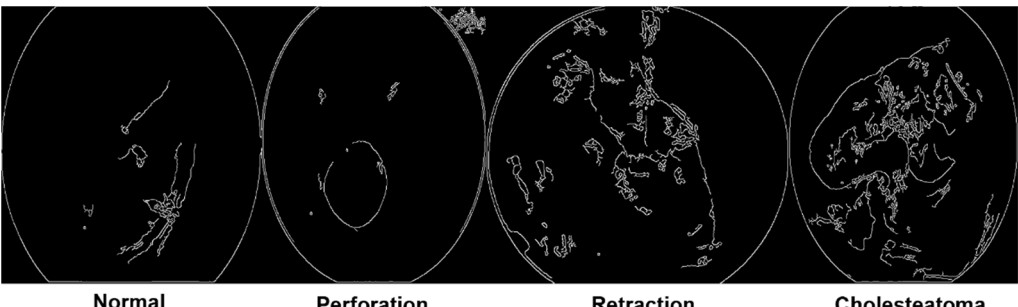

Normal  Perforation  Retraction  Cholesteatoma

**Figure 6.** Image after applying Canny edge detection.

### 3.3. Sobel Edge Detection

Sobel edge detection was used to compare the application of other edge-detection algorithms. The Sobel edge-detection algorithm is a method of calculating using $3 \times 3$ size matrices. The $3 \times 3$ matrix detects the amount of change by comparing the values before and after each direction relative to the center. We applied Gaussian filters twice to minimize noise, as in Canny edge detection. Sobel edge detection has a derivative method in the x direction and a derivative method in the y direction, which is sensitive to the amount of change in the x direction in the image and a derivative method in the y direction is sensitive to the amount of change in the y direction. Generally, the affected area is produced in the tympanic membrane and the tympanic membrane has a significant amount of change in the x direction. When we applied both the x-direction and y-direction methods, as in Figure 7, the tympanic membrane and affected area were more clearly detected by the x-direction

method. Therefore, we compared it with Canny edge detection using Sobel edge detection with an x-direction derivative. The comparisons are referenced in Section 6.

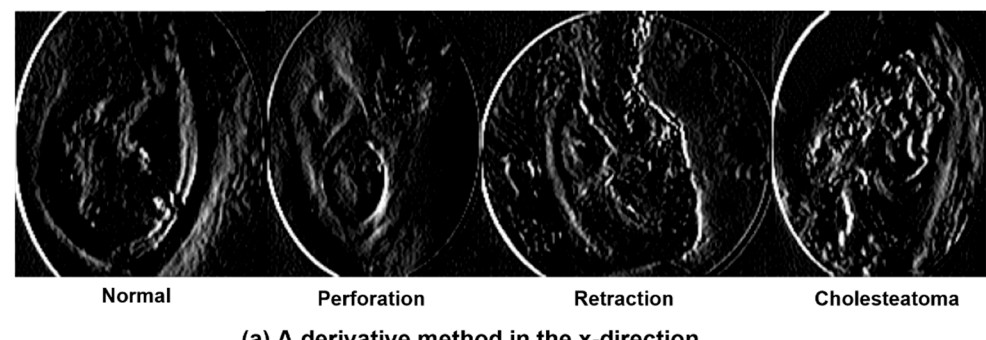

(a) A derivative method in the x-direction

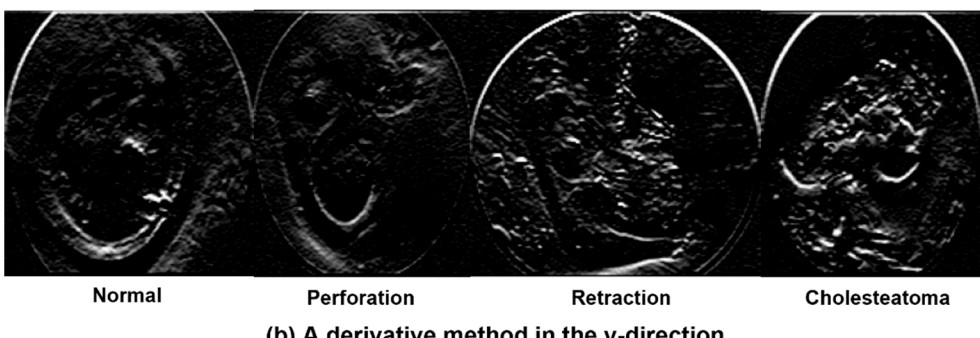

(b) A derivative method in the y-direction

**Figure 7.** Images according to the derivative direction of Sobel edge detection.

## 4. VAE

For use in real-world clinical circumstances, AI systems that leverage medical data should be able to discover and categorize a greater variety of instances rather than being prejudiced to certain instances. Several studies have been conducted on medical AI; however, most suffer from insolubility and deflection outside the training region, owing to the nature of supervised learning [11]. To overcome such challenges, we utilize a VAE for retrieving the abnormality scores. This section describes how abnormality scores can be extracted from the VAE.

The VAE assumes that a latent variable z influences data x and aims to identify z to develop a fresh data sample comparable to training data x like Figure 8. The autoencoder (AE) is extracted to a single-value latent variable for encoder learning. Moreover, the primary purpose of the VAE is to generate a decoder, which is more useful than AE in understanding the distribution of training data because latent variable x appears as a distribution that optimizes the likelihood of data x based on a Gaussian probability distribution.

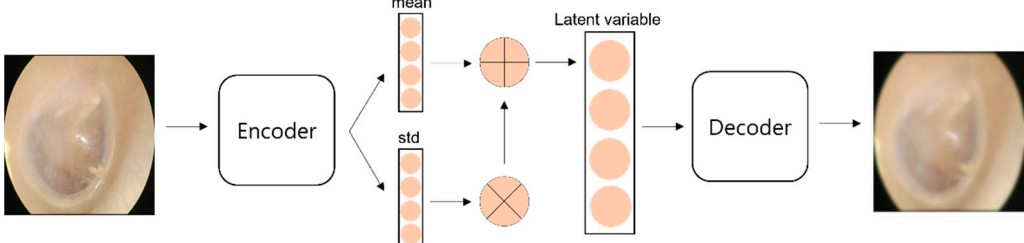

**Figure 8.** Architecture of VAE.

Initially, the VAE learns only from the normal tympanic membrane endoscopic images. Adam was used as the optimizer and the learning rate was learned by 100 epochs at $1 \times 10^{-3}$. The abnormality scores are calculated using the VAE that learned only the normal class using test data containing classes, such as perforation, retraction, and cholesteatoma like Figure 9. The abnormality scores are calculated using the reconstruction error and Kullback–Leibler diversity, which are calculated by factors resulting from differences in the distributions between the normal class and test data learned by the model. In other words, abnormality scores indicate the distance between the normal class and test data distributions in this study.

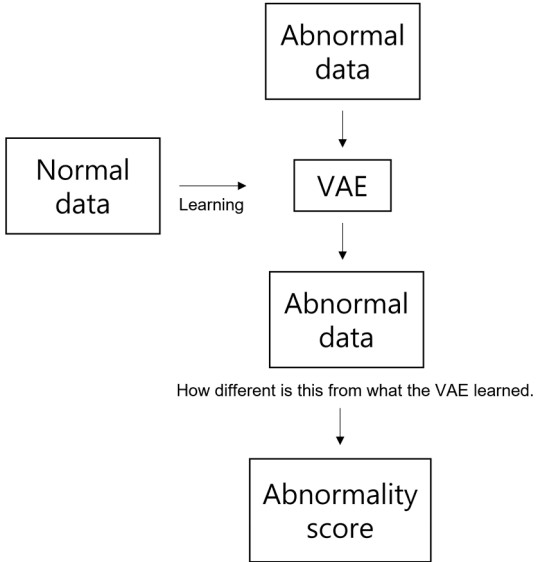

**Figure 9.** Abnormality score extraction process.

The VAE loss is determined as the sum of the abnormality scores: the reconstruction error and Kullback–Leibler diversity. The equations show the formula for computing the loss value of a VAE. The first item on the right is the Kullback–Leibler diversity and the second item is the reconstruction error.

$$\hat{\mathcal{L}}^{\mathrm{B}}\left(\theta, \phi; x^{(i)}\right) = -D_{KL}\left(q_\phi\left(z \middle| x^{(i)}\right) \parallel p_\theta(z)\right) + \frac{1}{L}\sum_{l=1}^{L}\left(\log p_{\theta\left(x^{(i)} \middle| z^{(i,l)}\right)}\right)$$

The Kullback–Leibler diversity is a function used to determine the difference between two probability distributions statistically. The VAE estimates the informative distance of $q\phi(z|x^{(i)})$, which is the probability distribution of $x$ that can be derived from $p\theta(z)$ and $z$. In other words, when the training data are distributed, the difference in entropy that can occur while sampling the input distribution of the test data is computed. The reconstruction error is the rate at which errors occur when the test data are reconstructed using a model. For instance, if data, such as perforation, retraction, and cholesteatoma, are fed into a model that learns only the normal class, the error rate increases during the reconstruction phase because a different distribution must be reconstructed from the learned data. The error rate that occurs during this process is known as the reconstruction error. We collected the reconstruction error and Kullback–Leibler diversity for the Section 3 dataset and applied them for detection.

## 5. Results

This section discusses the outcomes of anomaly detection using the scores outlined in Section 4. We executed the K-NN algorithm using the abnormality scores from Section 4.

The K-NN algorithm is a distance-based classification analysis model that categorizes data by referring to the labels of k pieces of data near the observation's Y value. As shown

in Figure 10, for instance, if the k value is set to 3, data are classified as orange; however, if it is set to 7, the data are classified as green. Similar to the K-NN technique, the k-means algorithm clusters data rather than classifying them. When analyzing the distribution of the abnormality scores, we determined that although the normal and abnormal groups are unclearly separated dichotomously, as shown in Figure 11, although the distribution of the normal and abnormal tympanic membrane scores was not clearly separated, it was confirmed that the distribution of the scores of each class were close. Therefore, applying the K-NN algorithm with neighboring data is more reasonable than clustering based on the relationships.

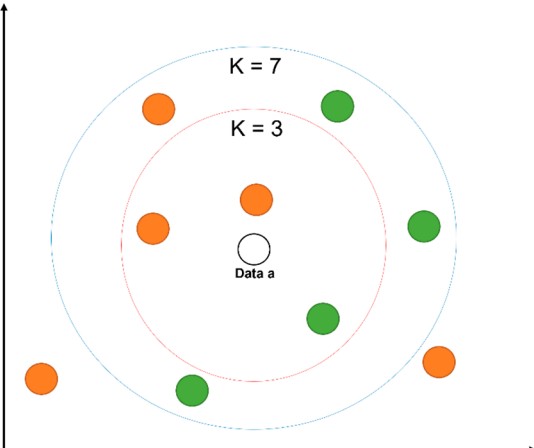

**Figure 10.** Classification results using K-nearest neighbor algorithm.

When we applied the K-NN algorithm to each data-specific abnormal datum and proceeded with the binary classification of normal and abnormal data, we detected abnormal data with an accuracy of 94.5% at k = 10, including anomalies that were difficult to identify for certain diseases and conditions.

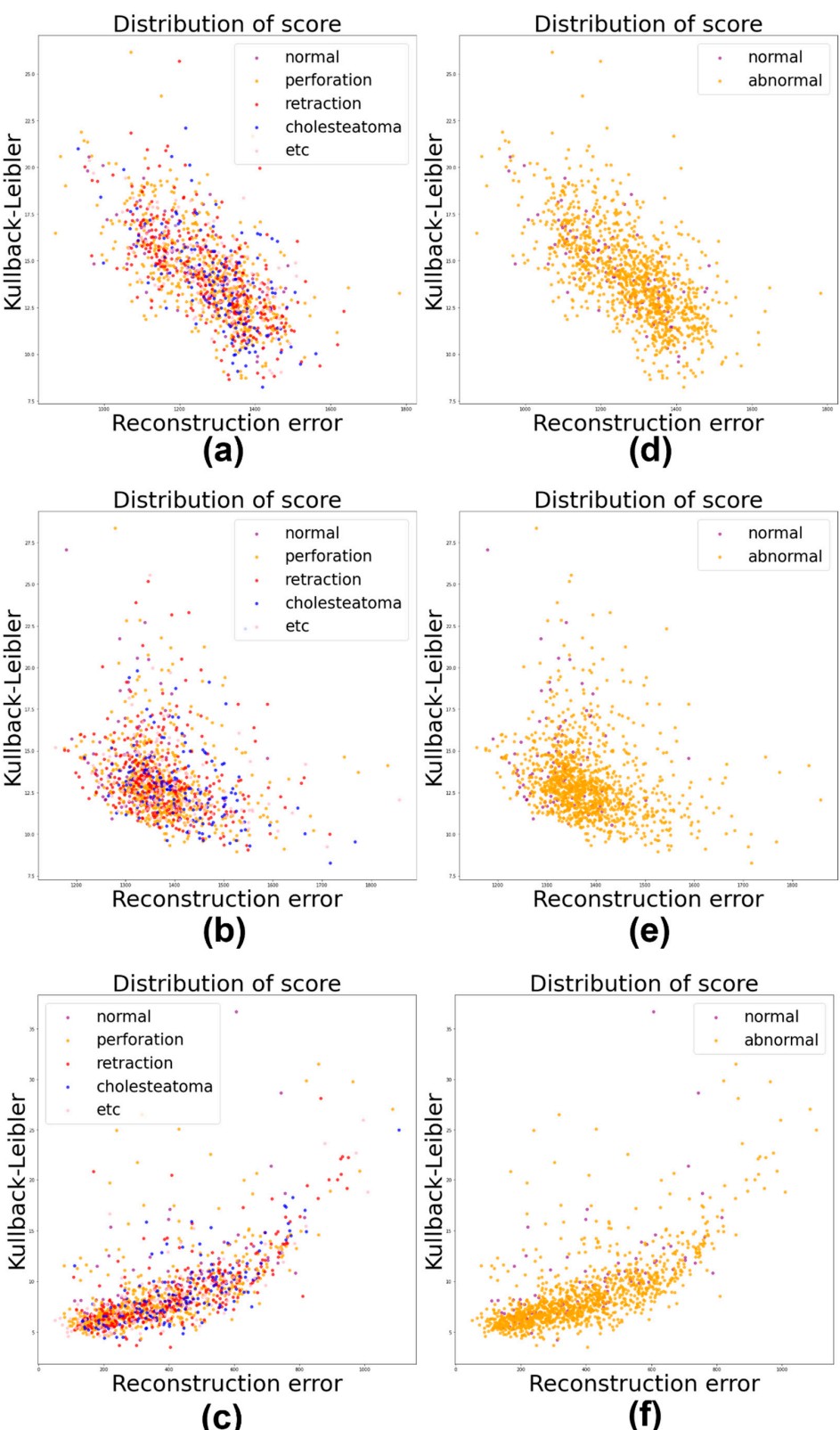

**Figure 11.** Distribution of abnormality scores: (**a**,**d**) are the original data; (**b**,**e**) are data after adaptive histogram equalization; and (**c**,**f**) are data after Canny edge detection. (**a**–**c**) are distributions for five classes: normal, performance, retraction, cholesteatoma, and classes that specialists struggle to classify as a particular disease but are certainly abnormal; and (**d**–**f**) are the distributions of normality/abnormality.

## 6. Discussion

Middle ear diseases are a disorder that may lead to potentially catastrophic side effects, including hearing loss and vestibular dysfunction [1]. They are particularly common in children and have been determined to have a relapse rate of over 80% [2]. Although the early detection and treatment of middle ear problems are essential, the average diagnostic accuracy of pediatricians and otolaryngologists in primary care is less than 70% [3]. Middle ear diseases are usually diagnosed by otoscopy; therefore, medical image processing, such as tympanic membrane endoscopy, is necessary for clinical analysis and data exploration [23]. The implementation of AI technology may be justifiable because such data are highly complex and may be difficult to visually interpret and diagnose from a single image [24].

In this study, we performed semantic decomposition to efficiently apply complicated tympanic membrane endoscopic images to the model. First, AHE was applied to maintain only structural information in the middle ear, except RGB values. Then, Canny edge detection was used to verify that only the edge of the affected region remained. The model learned structural information in the middle ear, information about the damaged edge, and original RGB data. The model then generated an abnormality score and applied the K-NN method to detect anomalies with an accuracy of 94.5%. As a result of checking the misclassified images, most of the misclassified cases were classified as abnormal despite normal tympanic membranes. For example, the misclassified cases were congested or not clean surface tympanic membranes, as shown in Figure 12.

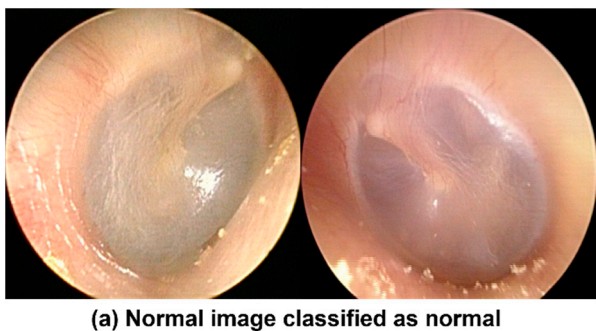

**(a) Normal image classified as normal**

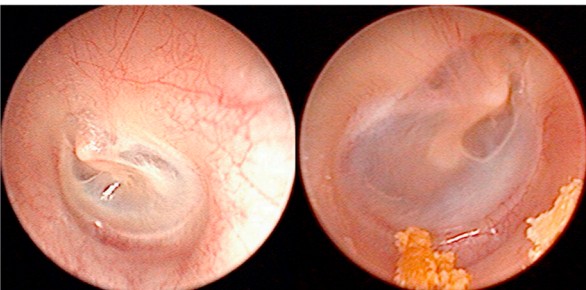

**(b) Normal image classified as abnormal**

**Figure 12.** Example of misclassified images.

We also performed comparisons with supervised learning models to validate our method. First, we adopted GoogLeNet as a supervised learning model for comparison. GoogLeNet trained four classes: normal tympanic membrane, the perforation of the tympanic membrane, retraction of the tympanic membrane, and cholesteatoma. The optimizer used Adam, the cross-entropy function utilized the loss function, and the learning rate was learned as $1 \times 10^{-4}$. The validation accuracy was 90% in epoch 25.

To verify the purpose of our study, we used test data that the model did not learn. The data used in the test did not train at both supervised and unsupervised learning models to verify the purpose of our study. The test images were not perforation, re-traction, and

cholesteatoma. It was difficult to diagnose specific diseases from the images but they were certainly abnormal.

The algorithm proposed in this paper classified all 103 abnormal tympanic mem-brane images as abnormal. However, the supervised learning model classified 25 tym-panic membrane images as normal. As shown in Table 1, the performance of the un-supervised learning model was higher for all indicators. Because the supervised learning model is hard to apply outside of the training area, on the other hand, our unsupervised-based method detected abnormal tympanic membranes with higher accuracy than supervised learning models, even though only normal tympanic membrane was learned. Comparisons with supervised learning models on the same data demonstrate that our method can diagnose more anomalies.

**Table 1.** Comparison of unsupervised and supervised learning methods.

| Methods | Evaluation Outcomes | | |
|---|---|---|---|
| | Accuracy | AUC | F1-Score |
| Unsupervised method | 94.5% | 0.92 | 0.96 |
| Supervised method | 80.4% | 0.80 | 0.77 |

In addition, to verify the semantic decomposition approaches, we performed compari-son experiments. The experiments use unsupervised learning models identically and apply preprocessing methods differently. When using only original data without AHE and Canny edge detection, abnormal tympanic membranes were detected with an accuracy of about 2–3% lower than our method.

Furthermore, we conducted an experiment that applied Sobel edge detection instead of Canny edge detection and the other preprocessing method was the same as ours.

Images with original data and AHE and images with Sobel edge detection instead of Canny edge detection were shown to be 2–3% less accurate than our method. In learn-ing VAE, Canny edge detection had a 25% lesser loss rate than Sobel edge detection. Our method exhibited higher performance than other image-processing methods in un-supervised learning methods as well as supervised learning methods. The results are summarized in Table 2.

**Table 2.** Anomaly detection performance by pre-processing method.

| Methods | Evaluation Outcomes | |
|---|---|---|
| | Accuracy | AUC |
| Canny edge detection | 94.5% | 0.92 |
| Sobel edge detection | 91.9% | 0.91 |
| Original image | 91% | 0.87 |

In this paper, the proposed method can identify even the tympanic membrane that is ambiguous to diagnose as a particular disease but is certainly abnormal and it used only normal data for model training. This shows the applicability in real-world clinics where various scenarios and difficulties in data collection appear. Our method exhibited higher performance than other image-processing methods in unsupervised learning methods as well as supervised learning methods.

## 7. Conclusions

We decomposed high-complexity tympanic membrane endoscopic images based on clinical features and performed the task of identifying various abnormal data not reflected in the algorithm. In the existing otolaryngology, the only application of AI technology

was supervised learning. Supervised-learning-based diagnostics are difficult to use in real-world clinical situations because it is hard for the model to diagnose unlearned data. Consequently, we performed the task of detecting anomalies in tympanic membrane endoscopic images using unsupervised learning models, VAE. Unsupervised learning models find it difficult to learn complex data, such as tympanic membrane endoscopic images. Therefore, we first worked on semantically decomposing them based on clinical features. The AHE and Canny edge-detection algorithms were applied. Further, we made three kinds of tympanic membrane endoscopic images as the original image, an image highlighting only the transparency and depth of the tympanic membrane, and an image that left only the boundary of the affected area. After image processing, only normal tympanic membranes were trained in the VAE. The VAE calculated the degree to which the image distribution of the abnormal tympanic membrane differs from that of the normal tympanic membrane as an abnormality score. By applying the K-NN algorithm to the calculated abnormality score, we were able to detect anomalies with an accuracy of 94.5%. Compared to supervised learning models with test data that were not used as learning data, our method was able to detect more anomalies than supervised learning methods. In addition, we obtained higher accuracy than the results of using only the original images. Therefore, we expect our method to be easier than the existing methods applied to otolaryngology in real clinical situations. In addition, the proposed method may be a solution to the chasm phenomena in medical AI, such as the difficulties in database development and highly complex data. Future medical AI research should involve attempts to obtain clinically relevant outcomes for patients in real-world practice settings instead of simply enhancing the classification accuracy of a model based on technological advancements.

**Author Contributions:** Conceptualization, D.S., I.S.S., J.K., J.C. and Y.L.; methodology, D.S. and Y.L.; software, D.S. and Y.L.; resources, I.S.S., J.K. and J.C.; data curation, I.S.S. and J.C.; writing—original draft preparation, D.S. and Y.L.; writing—review and editing, I.S.S., J.C. and Y.L.; supervision, I.S.S., J.K., J.C. and Y.L.; project administration, D.S. and Y.L. All authors have read and agreed to the published version of the manuscript.

**Funding:** This study was supported by a Korea University Grant (K2210761) and the MSIT (Ministry of Science and ICT), Korea, under the ICAN (ICT Challenge and Advanced Network of HRD) pro-gram (IITP-2022-RS-2022-00156439) supervised by the IITP (Institute of Information & Communications Technology Planning & Evaluation). This work was supported by the Institute of Information & communications Technology Planning & Evaluation (IITP) grant funded by the Korea government (MSIT) (No. RS-2022-00155885, Artificial Intelligence Convergence Innovation Human Resources Development (Hanyang University ERICA)), (No. 2020-0-01343, Artificial Intelligence Convergence Research Center (Hanyang University ERICA)).

**Institutional Review Board Statement:** The study was conducted in accordance with the Declaration of Helsinki and approved by the Institutional Review Board of Korea University Ansan Hospital (protocol code 2021AS0329 and date of approval 2 November 2021).

**Informed Consent Statement:** Informed consent in this study was waived by the institutional review board of the Korea University Ansan Hospital because it is a retrospective study.

**Data Availability Statement:** Not applicable.

**Conflicts of Interest:** The authors declare no conflict of interest.

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
