# Peer review of "Semantic Decomposition and Anomaly Detection of Tympanic Membrane Endoscopic Images"

_applsci, doi:10.3390/app122211677_

Round 1
Reviewer 1 Report
This article is of great practical significance, but the following points still need to be modified:
1.The statement in the abstract does not match the actual algorithm。
“Therefore, we propose that unsupervised learning-based anomaly detection of the tympanic membrane can solve the limitations of existing supervised learning methods.”
In general, we believe that data with labels should not be input in unsupervised learning, and you have input normal classes。
2.The experimental process should be more detailed, for example, the comparison of different methods on the same data set, the influence of hyperparameters on accuracy, etc.
3.In the results of the paper, it is mentioned that the accuracy of the abnormal data is 94.5,“including anomalies that were difficult to identify for certain diseases and conditions.” (line144).If it is a human judgment, there is an objection to the content of the 16 lines, because the supervised model can also manually subdivide the pathology class after network classification. If it can be reflected in the model, it should be clearly indicated.
Author Response
Please see the attachment. Thank you:)

Reviewer 2 Report
In this paper, a method is propose for anomaly detection of tympanic membrane endoscopic images. A VAE is applied after the semantic decomposition of high-complexity data. The novelty is less. The experiment is too easy. The authors do not make comparison with other methods. Also, they do not analyze the improvements that they make. They should respectively demonstrate that each improvement is efficient. Therefore, I would reject the paper.
Updated more comments 10.28
In this paper, a method is proposed for anomaly detection of tympanic membrane endoscopic images. A VAE is applied after the semantic decomposition of high-complexity data. The novelty is less and the experiment is too easy. The major revisions are as follows:
(1) The authors do not make comparison with other methods.
(2) They do not analyze the improvements that they make. They should respectively demonstrate that each improvement is efficient.
(3) The method includes a little of innovative work. It just integrates the exiting methods without any changes.
(4) Some sentences are hard to understand, e.g., line 162, 163.
(5) The fontsize of Figure 10 is small.

Reviewer 3 Report
The paper proposes a novel approach for detecting abnormalities in tympanic membrane endoscopic images using an unsupervised learning model VAE, which helps to overcome the limitations in supervised learning models, such as the inapplicability and bias in real-life situations.
The paper is well-organized. "Abstract," "Introduction," and "Materials and Methods" are very well written. Especially the data preprocessing is described clearly and in detail.
However, the remaining sections are incomplete and very short. The authors could not represent their results properly. For instance,
1) Section 5 does not discuss the results related to VAE,
2) the achievement of the highest accuracy at k=10 is not convincing,
3) the distribution of abnormality scores is only discussed in the figures,
4) Section 6 is short and does not discuss the obtained results in detail and the benchmarking provided sounds too informal and like conclusions,
5) Section 7 is also short and does not adequately summarize the research.
I believe that the authors have done great work. The obtained results are correct and valuable. They need to revise their paper to improve its presentability and readability.
Round 2
Reviewer 1 Report
This paper has profound practical significance and application value, and thanks to the contributions made by relevant personnel. There is still much room for improvement in the details and method demonstration of this paper, and I hope to do better in the future scientific research.
Reviewer 2 Report
The revisions have been made according to the comments
Reviewer 3 Report
The authors have made significant improvements to the paper and adequately responded to the comments and suggestions. I would like to recommend accepting the paper with minor English spell-checking.